# Feature Extraction of Ship-Radiated Noise Based on Regenerated Phase-Shifted Sinusoid-Assisted EMD, Mutual Information, and Differential Symbolic Entropy

**DOI:** 10.3390/e21020176

**Published:** 2019-02-14

**Authors:** Guohui Li, Zhichao Yang, Hong Yang

**Affiliations:** School of Electronic Engineering, Xi’an University of Posts and Telecommunications, Xi’an 710121, China

**Keywords:** ship-radiated noise, regenerated phase-shifted sinusoid-assisted EMD, mutual information, differential symbolic entropy, feature extraction

## Abstract

To improve the recognition accuracy of ship-radiated noise, a feature extraction method based on regenerated phase-shifted sinusoid-assisted empirical mode decomposition (RPSEMD), mutual information (MI), and differential symbolic entropy (DSE) is proposed in this paper. RPSEMD is an improved empirical mode decomposition (EMD) that alleviates the mode mixing problem of EMD. DSE is a new tool to quantify the complexity of nonlinear time series. It not only has high computational efficiency, but also can measure the nonlinear complexity of short time series. Firstly, the ship-radiated noise is decomposed into a series of intrinsic mode functions (IMFs) by RPSEMD, and the DSE of each IMF is calculated. Then, the MI between each IMF and the original signal is calculated; the sum of MIs is taken as the denominator; and each normalized MI (norMI) is obtained. Finally, each norMI is used as the weight coefficient to weight the corresponding DSE, and the weighted DSE (WDSE) is obtained. The WDSEs are sent into the support vector machine (SVM) classifier to classify and recognize three types of ship-radiated noise. The experimental results demonstrate that the recognition rate of the proposed method reaches 98.3333%. Consequently, the proposed WDSE method can effectively achieve the classification of ships.

## 1. Introduction

By analyzing ship-radiated noise, a slice of crucial information such as the type, speed, and tonnage of the ship can be extracted. Consequently, the identification of ship-radiated noise plays a significant role in the military and economic fields [1,2,3]. Due to the influence of marine environment noise, the actual measured ship-radiated noise is non-Gaussian, non-stationary, and non-linear [4,5,6]. Therefore, it is an arduous task to identify ship-radiated noise effectively. To realize the feature extraction of ship-radiated noise, scholars have proposed Fourier transform, wavelet transform and modern spectrum estimation [7]. However, these methods have some limitations. As an example, Fourier transform analysis cannot reflect the time-varying characteristics of the signal well; the wavelet transform needs to select the wavelet basis function and the decomposition level, in advance [8]. Consequently, it is indispensable to find a feature extraction method with higher recognition accuracy.

Empirical mode decomposition (EMD) [9] is an adaptive signal decomposition algorithm, which can decompose the original signal into a series of intrinsic mode functions (IMFs) with different frequencies. Therefore, it provides a new idea for underwater acoustic signal processing. However, EMD may cause the mode mixing problem, which leads to different frequency components in an IMF. To overcome the mode mixing problem, Wu et al. [10] proposed ensemble empirical mode decomposition (EEMD), which uses the statistical properties of white noise to reduce the mode mixing to some extent. EEMD has some positive effects on signal decomposition. However, the ensemble size of EEMD is required to be very large, resulting in heavy computation. In addition, if the amplitude and the iteration number of the added white noise are not appropriate, a sea of undesirable components will appear in the results of decomposition. Regenerated phase-shifted sinusoid-assisted EMD (RPSEMD) [11] is intended to solve the mode mixing problem by designing sinusoids and a high-performance phase-shifting scheme. The original signal is decomposed by RPSEMD to solve the mode mixing problem and large computation costs [12].

Shannon entropy is a measure of uncertainty or irregularity. So far, various entropies have been developed, including sample entropy (SE) [13], permutation entropy (PE) [14], symbol entropy (SymEn) [15], and so on. Entropy-based algorithms have been applied in a variety of fields, but these entropies have some limitations. As an example, SE is powerful, but it is not fast enough for large datasets. PE is conceptually simple and computationally fast, and the method does not consider the mean value of amplitudes and differences between amplitude values [16]. Differential symbolic entropy (DSE) [17,18] recently proposed by Yao et al. is a new version of symbol entropy. It extracts local nonlinear dynamic information from three adjacent elements and uses an adjustable controlling parameter to improve flexibility in nonlinear complexity detections.

In recent years, scholars have applied the mode decomposition algorithm and entropy to various signal feature extraction fields and achieved promising results. Yang et al. [19] drew the statistical center frequency of the spectrum of IMFs as new line spectrum characteristic of the underwater acoustic signal and adopted the law of the nearest neighborhood to recognize it. Bao et al. [20] proposed a modified EMD technique by estimating the local mean of a signal via a windowed average, and they used this new tool to achieve the adaptive and efficient extraction of modulated cavitation noise from ship-radiated noise. Li et al. [21] used EMD to decompose the ship-radiated noise, taking the PE of the IMF with the highest energy as the feature vector. Chen et al. [6] proposed a feature extraction method of ship-radiated noise based on hierarchical cosine similar entropy. Compared with multi-scale sample entropy, the classification accuracy was significantly improved from 75%–95.63%. Zhou et al. [22] proposed a novel bearing multi-fault diagnosis approach based on EEMD, weighted permutation entropy, and support vector machine (SVM). The results demonstrated that this method can effectively detect bearing faults and maintain a high accuracy rate of fault recognition when a small number of training samples is available. Bao et al. [23] proposed a ship classification method based on analysis of ship-radiated noise in subspaces of IMFs. This substantiated that ship-radiated noises contain components with deterministic nonlinear features serving the efficient classification of ships well. Gan et al. [24] proposed a rolling bearing fault diagnosis approach based on composite multiscale weighted permutation entropy, joint mutual information (JMI), and the k-nearest-neighbor (KNN) classifier. The analysis results indicated that this algorithm could effectively identify different fault rolling bearing conditions. Knobles et al. [25] used the multistep maximum entropy (ME) method to quantify the conditional probability distributions for a space consisting of both ocean waveguide parameters and aspect-dependent source levels. Yang et al. [26] used EEMD to decompose the ship-radiated noise, and the energy difference between the high and low frequency was extracted and analyzed. The results showed that the energy difference between the high and low frequency is at the same level for similar ships, but there is an obvious difference for different types of ships. Fırat et al. [27] proposed a scheme for compressive sensing of propeller tonals, and the results showed that the spectral correlation function of cyclostationary propeller noise is sparse, and a linear relationship between the compressive and Nyquist-rate cyclic modulation spectra was derived.

In this paper, a feature extraction method of ship-radiated noise based on RPSEMD, mutual information (MI), and DSE is proposed. Firstly, RPSEMD is used to decompose the ship-radiated noise into a series of IMFs. Then, the MI between each IMF and the original ship-radiated noise is calculated, and the normalized MI is used to weight the corresponding IMF’s DSE to obtain the weighted DSE (WDSE). Finally, the feature vector WDSE is put into SVM for classification.

## 2. Basic Theory

### 2.1. Regenerated Phase-Shifted Sinusoid-Assisted EMD

#### 2.1.1. The Traditional EMD Algorithm

The specific steps of EMD are summarized as follows [9]:Step 1.Connect the local maxima/minima of the original signal *x*(*t*) to obtain the upper/lower envelope using the cubic spline.Step 2.Derive the local mean of envelope, *m*(*t*), by averaging the upper and lower envelopes.Step 3.Extract the temporary local oscillation d(t)=x(t)−m(t).Step 4.If d(t) satisfies some predefined stoppage criteria, d(t) is assigned as an IMF denoted as *c_m_*(*t*) where *m* is the IMF index. Otherwise, set x(t)=d(t), and repeat Steps 1–3.Step 5.Compute the residue *r_m_*(*t*) = *x*(*t*) − *c_m_*(*t*).Step 6.Set *x*(*t*) = *r_m_*(*t*), and repeat Steps 1–5 to extract the next IMF. The final residue is denoted as r(t).

The result of EMD can be expressed as:(1)$$x(t)=\sum_{k=1}^{K}c_{k}(t)+r(t)$$
where *k* is the index and *K* denotes the total number of IMFs. An ideal IMF should only include one monocomponent, such as IMF(t)=a(t)cos(2πf(t)t) [9], where a(t) and f(t) are the instantaneous amplitude and instantaneous phase, respectively. However, in fact, the mode mixing problem often appears, resulting in an IMF being split into two adjacent IMFs or different IMFs existing in an IMF. Then, an IMF containing *J* IMFs will follow the form:(2)$$IMF(t)=\sum_{j=1}^{J}a_{j}(t)cos\left(2\pi f_{j}(t)t\right)$$

The work in [28] indicated that a two-mode signal, with $a_{1}$, $f_{1}$ and $a_{2}$, $f_{2}$ as their respective amplitudes and frequencies, can be separated only if the cutoff frequency ratio is no more than 0.67, which can be expressed as:(3)$$f_{1}/f_{2}>1.5\;\mathrm{or}\;f_{2}/f_{1}<0.67,\;(f_{1}>f_{2})$$

Therefore, we consider that two points at $t_{1}$ and $t_{2}$ belong to the same IMF if $f(t_{1})/f(t_{2})\in [0.67,1.5]$. Further, the extreme point detected in the sifting of EMD will all belong to the high-frequency mode if:(4)$$a_{1}f_{1}>a_{2}f_{2},\;(f_{1}>f_{2})$$

RPSEMD uses Equations (3) and (4) as the criteria for separating two IMFs. In view of the cause of the mode mixing problem, making the extreme points of each IMF distribute evenly becomes an important factor to solve the mode mixing problem, which can be achieved by adding auxiliary signals.

#### 2.1.2. The Main Idea of RPSEMD

The novelty of RPSEMD can be summarized into three aspects. First, a general-form sinusoid is used as the auxiliary signal:(5)$$s_{k}(t|a_{k},f_{k},\theta_{k})=a_{k}cos\left(2\pi f_{k}t+\theta_{k}\right)$$
where *k* represents the *k*^th^ stage of decomposition to extract the IMF $c_{k}(t)$; $a_{k}$, $f_{k}$, and $\theta_{k}$ denote the amplitude, frequency, and phase, respectively. Second, the sinusoid-assisted signal $s_{k}(t)$ is designed based on the IMF selected after IMF clustering analysis. This active way ensures that the RPSEMD is deterministic and is able to solve the mode mixing problem more efficiently. Third, $s_{k}(t)$ is shifted by $\theta_{k}$ to change the positions of the extreme point, not only helping to retain more details of the separated IMFs, but also ensuring $s_{k}(t)$ is completely canceled out in the final results. The specific steps of RPSEMD are summarized as follows [11]:Step 1.Initialize k=1.Step 2.Apply EMD to x(t) and then determine $a_{k}$ and $f_{k}$ with the resulting IMFs. $\theta_{ki}$ is acquired by uniformly sampling in [0,2π] with the phase shifting number I(1≤i≤I). After this, $s_{k}(t|a_{k},f_{k},\theta_{ki})$ is obtained.Step 3.The EMD of $x(t)+s_{k}(t|a_{k},f_{k},\theta_{ki})$ is performed, which aims to obtain the first IMF. The final IMF $c_{k}(t)$ is calculated by averaging all these first IMFs.Step 4.Remove $c_{k}(t)$ from $x(t)\leftarrow x(t)-c_{k}(t)$. Let k=k+1.Step 5.Repeat Steps 2–4 until no more IMF can be obtained. Consequently, the final x(t) is regarded as a residue r(t).

#### 2.1.3. Selecting the Parameters of $s_{k}(t)$

The mode mixing problem occurs because the extreme points of the IMF in the signal are unevenly distributed. Therefore, the target of adding $s_{k}(t)$ is to imitate a homogenous IMF.

(1) Determining $a_{k}$ and $f_{k}$ of $s_{k}(t)$: As described above, the first task is to find the target IMF to be imitated. Equation (2) shows that an IMF is generally a composition of *J* IMFs if the mode mixing problem exists. In this sense, the initial IMFs by EMD, denoted as $ci_{k^{\prime }}(t)$ ($1\leq k^{\prime }\leq K^{\prime }$), can be used as a reference to obtain the target IMF.

a. Obtain the target IMF to be imitated: Since EMD sifts a signal from high-frequency to low-frequency, we analyze the IMFs of $ci_{1}(t)$ and pick out the one with the highest frequency as the target IMF. However, x(t) is unknown, and the IMFs of $ci_{1}(t)$ are quite arbitrary, while the only certainty is that the frequencies of different IMFs are constrained by Equation (3). In view of these issues, clustering analysis is introduced to cluster the instantaneous frequencies of the extreme in $ci_{1}(t)$, to classify the IMFs of $ci_{1}(t)$. Wang et al. [11] used hierarchical clustering since it is a nonparametric approach. Specifically, based on the instantaneous frequencies of the extreme, the hierarchical clustering is executed as follows: first, the Euclidean distances between instantaneous frequencies are calculated, then a tree of hierarchical clusters with these distances is created, and finally, the clusters are constructed.

As the number of the clusters *P* is unknown, we start with $P=K^{\prime }$ to ensure P≥J and iteratively reduce *P* by $P^{m}$, until $P^{m}$ is zero. Finally, the clusters whose mean frequencies do not satisfy Equation (3) are excluded when comparing with that of $s_{k-1}(t)$ obtained in the (k−1)^th^ stage of decomposition.

b. Determining $a_{k}$ and $f_{k}$: The final *P* clusters are treated as *P* IMFs with amplitudes $ac_{p}(1\leq p\leq P)$ and mean frequencies $fc_{p}$, and the one with the highest frequency, say the $p^{0}$^th^ cluster, is determined as the target IMF. We then set $f_{k}=fc_{p^{0}}$ and $a_{k}=ac_{p^{0}}$.

Two adjustments of $a_{k}$ are executed based on Equations (3) and (4) to ensure the added $s_{k}(t)$ can be separated from the IMFs that are much easier to be mixed with $s_{k}(t)$, which are described as follows.

First, to separate from the other IMFs in $ci_{1}(t)$, the current $a_{k}$ is increased to $max(a_{k},\;max(ac_{p}\times fc_{p}/f_{k}))$ according to Equation (4). Second, to separate all IMFs in $ci_{2}(t)$, $a_{k}$ is further adjusted as $a_{k}+a_{k}^{*}$ if $a_{k}$, $f_{k}$, and $a_{k}^{*}$, $f_{k}^{*}$ do not satisfy Equations (3) and (4), where $a_{k}^{*}$ and $f_{k}^{*}$ are obtained using the above procedure of obtaining $a_{k}$ and $f_{k}$ for $ci_{2}(t)$. This adjustment means merging the target IMF because it is judged to be split into $ci_{1}(t)$ and $ci_{2}(t)$.

The algorithm of determining $a_{k}$ and $f_{k}$ is summarized below:Step 1.For the extreme of $ci_{1}(t)$, get their instantaneous amplitudes $ai_{1}(e)$ and instantaneous frequencies $fi_{1}(e)$, where *e* indicates the index of an extreme.Step 2.Repeatedly classify $fi_{1}(e)$ into *P* clusters by adjusting $P\leftarrow P-P^{m}$ until any two clusters satisfy Equation (3).Step 3.Find the $p^{0}$^th^ cluster and set $f_{k}=fc_{p^{0}}$ and $a_{k}=ac_{p^{0}}$.Step 4.Adjust $a_{k}$ to ensure $s_{k}(t)$ can be separated from the IMFs clustered in $ci_{1}(t)$ and $ci_{2}(t)$.

(2) Determining $\theta_{ki}$ of $s_{k}(t)$: Including the phase $\theta_{ki}$ in $s_{k}(t)$ has two intensions: first, make that the auxiliary sinusoid be added complementarily in pairs to make sure they can be eliminated ultimately; second, change the relative positions of the extreme between the added sinusoid and the signal to retain largely the details of the separated IMFs. The second intension is the response to the fact that, occasionally, some extreme describing the details of an IMF may be hidden and cannot be detected after adding a sinusoid; while shifting the sinusoid by a phase can make those extremes appear.

The solution of shifted phase $\theta_{k}$ can be defined as follows:(6)$$\theta_{ki}=\frac{2\pi }{N}i,\;0\leq i\leq N-1$$
where *N* denotes the number of shifts. To achieve the first intention, *N* must be an even number so that $\theta_{ki+N/2}=\theta_{ki}+\pi (0\leq i\leq N/2)$ and thus $s_{ki+N/2}(t)=-s_{ki}(t)$. For the second intention, we set $N=2R(1/2f_{1})$ where $f_{1}$ comes from $s_{1}(t)$ in the first stage of decomposition and R(⋅) is a rounding operation. We select $f_{1}$ to determine *N* to balance the calculation and the ability of retaining the details of IMFs, because usually, the details of a high-frequency IMF are much easier to lose. The link of the RPSEMD code is: http://www3.ntu.edu.sg/home/mkmqian/RPSEMD.htm.

### 2.2. Differential Symbolic Entropy

#### 2.2.1. Traditional Symbolization

Symbolization plays an important role in symbolic dynamic analysis. The symbolic procedure inevitably leads to the loss of part of statistical information; however, it simplifies time series analysis and contributes to dynamic complexity detection by extracting symbolic dynamic information.

A symbolization in the works of Kurths et al. [29], using typical local dynamic symbolization, conducts symbolic transformation by comparing relationships between adjacent symbols. Given univariate time series $Y=\left\{y_{i},i=1,\ldots ,N\right\}$, traditional symbolization, being described as especially reflecting the dynamical properties of the record [29], transforms time series into a symbol sequence as Equation (7):(7)$$S_{i}(y_{i})=\left\{ \begin{array}{l} 0:\Delta y\geq 1.5\sigma_{\Delta }\\ 1:\Delta y>0\;\mathrm{and}\;\Delta y\leq 1.5\sigma_{\Delta }\\ 2:\Delta y>-1.5\sigma_{\Delta }\;\mathrm{and}\;\Delta y<0\;\\ 3:\Delta y\leq -1.5\sigma_{\Delta } \end{array}\right.$$

Traditional symbolic transformation refines the differences between neighboring elements, but it only considers two adjacent values and lacks flexibility due to the fixed $1.5\sigma_{\Delta }$.

#### 2.2.2. Differential Symbolization

Taking the relationships of three consecutive elements into account, Yao et al. [17,18] proposed differential symbolic transformation with a flexible controlling parameter. The complexity detection of this symbolization is attributed to detailed local dynamic information. Considering univariate time series $Y=\left\{y_{i},i=1,\ldots ,N\right\}$, the differences between the current element and its forward and backward ones are expressed as $D_{1}=\left\|y(i)-y(i-1)\right\|$ and $D_{2}=\left\|y(i+1)-y(i)\right\|$. The four-symbol differential symbolization with the controlling parameter *α* is obtained by the following formula:(8)$$S_{i}(y_{i})=\left\{ \begin{array}{l} 0:diff\geq \alpha \cdot var\\ 1:0\leq diff<\alpha \cdot var\\ 2:-\alpha \cdot var<diff<0\\ 3:diff\leq -\alpha \cdot var \end{array}\right.$$
where $diff=D_{1}-D_{2}$ and $var=\sqrt{(D_{1}^{2}+D_{2}^{2})/2}$. The symbolization in Equation (8) takes advantage of more detailed local information of complexity measures than traditional symbolic transformation.

Construction of symbol sequences, or words, is the next step by collecting groups of symbols together in temporal order. This coding process is to create symbol templates or words with finite symbols and has some similarities to embedding theory for phase space construction [18]. The symbol sequence will be coded into *m*-bit series C(i), and there are $4^{m}$ symbols in coded series considering the four-symbol differential symbolization. Taking 3-bit coding as an example, the code for ‘αβγ’ can be $c(i)=\alpha \cdot n^{2}+\beta \cdot n+\gamma$, where n=4. The procedure of symbolization and coding is illustrated in Figure 1. The probability of each code symbol is $P(\pi )=[p(\pi_{1}),p(\pi_{2}),\ldots ,p(\pi_{4^{m}})]$.

Finally, DSE is obtained by computing the Shannon entropy from the probability distribution for all the words, and then, it is normalized by its highest value, i.e., $\log_{2}4^{m}$, such that:(9)$$DSE=-\frac{1}{\log_{2}4^{m}}\sum p(\pi_{i})\log_{2}p(\pi_{i}),\;\mathrm{where}\;p(\pi_{i})\neq 0$$

### 2.3. Mutual Information

Originating from the classic and profoundly influential work by Shannon, the MI between discrete random variables *X* and *Y* is defined as [30]:(10)$$MI(X;Y)=\sum_{y\in Y}\sum_{x\in X}p(x,y)\log(\frac{p(x,y)}{p(x)p(y)})$$
where p(x,y) is the joint probability distribution function of *x* and *y*, p(x) and p(y) are the marginal probability distribution functions of *x* and *y*, respectively.

The MI of continuous random variables can be expressed as a double integral [31]:(11)$$MI(X;Y)=\int_{Y}\int_{X}p(x,y)\log(\frac{p(x,y)}{p(x)p(y)})dxdy$$

In probability theory and information theory, the mutual information of two random variables represents a measure of the interdependence of variables. If *X* and *Y* are independent, MI(X;Y)=0.

Furthermore, the MI can also be expressed as [31]:(12)$$\begin{array}{ll} MI(X;Y) & =H(X)-H(X|Y)\\ & =H(Y)-H(Y|X)\\ & =H(X)+H(Y)-H(X,Y)\\ & =H(X,Y)-H(X|Y)-H(Y|X) \end{array}$$
where H(X) and H(Y) are information entropy, H(X|Y) and H(Y|X) are conditional entropy, and H(X,Y) is the joint entropy of *X* and *Y*.

Assume that the original signal x(t) is decomposed by RPSEMD to obtain *K* IMFs (for convenience, the residue r(t) is counted as the last IMF), which are denoted as $IMF_{1}(t)$, $IMF_{2}(t)$, …, $IMF_{i}(t)$, …, $IMF_{K}(t)$, respectively. Therefore, the normalized MI (norMI) between the *i*^th^ IMF and the original signal can be expressed as:(13)$$norMI(IMF_{i};y(t))=\frac{MI(IMF_{i};y(t))}{\sum_{m=1}^{K}MI(IMF_{m};y(t))},\;i=1,2,\ldots ,K$$

For convenience, the norMI used in the following test represents the norMI between the IMF and the original signal, unless otherwise specified.

## 3. The Proposed Feature Extraction Method

The flowchart of the proposed feature extraction method is shown in Figure 2.Step 1.The three types of recorded ship-radiated noise are normalized.Step 2.The ship-radiated noise is decomposed into a series of IMFs by RPSEMD.Step 3.Calculate the DSE of each IMF.Step 4.The MIs between each IMF and the original signal are calculated, and then, the sum of all MIs is used as the denominator to calculate the normalized value of each MI, expressed as norMI.Step 5.The norMI is used as the weight coefficient to weight the corresponding DSE, and the feature vector WDSE is obtained.Step 6.The feature vector WDSE is input into the support vector machine for classification.

## 4. Analysis of the Simulation Signal

### 4.1. Performance Analysis of EMD, EEMD, and RPSEMD

To verify that RPSEMD can alleviate the mode mixing problem, we give an example here. The simulation signals are as follows:(14)$$s1=\left\{ \begin{array}{l} 0,\;0\leq n<0.25\\ 0.2\cos(300\pi n),\;0.25\leq n<0.35\\ 0,\;0.35\leq n<0.65\\ 0.2\cos(300\pi n),\;0.65\leq n<0.75\\ 0,\;0.75\leq n\leq 1 \end{array}\right.$$
(15)s2=0.3cos(50πn), 0≤n≤1
(16)s3=0.6cos(10πn), 0≤n≤1
(17)S=s1+s2+s3
where s1, s2, and s3 represent the three components of S, and the sampling frequency is 10 kHz. EMD, EEMD, and PRSEMD are used to decompose S. The simulation signals and the decomposition results are presented in Figure 3. According to the literature [26], the amplitude of the noise is set as 0.3, and the ensemble size is 100 for the EEMD method. It can be seen from Figure 3b that the IMF1 and IMF2 of the EMD have obvious mode mixing. EEMD alleviates the mode mixing to a certain extent, but the original signal is decomposed into nine mode components, of which IMF6–IMF9 have no practical physical meaning. Compared with EEMD, there is no need to select parameters for RPSEMD, so it has better practicability. Figure 3d manifests that the original signal is decomposed into three IMFs by PRSEMD, and these IMFs are very close to the three components of the original signal. Consequently, compared with EMD and EEMD, RPSEMD better optimizes the mode mixing problem. Further, the energy of the signal can be calculated by:(18)$$E=\sum_{i=1}^{N}p_{i}{}^{2}$$
where *N* represents the length of the signal and $p_{i}$ denotes the amplitude of the *i*^th^ sample point. The energy of simulation signals and IMFs is shown in Table 1 and Table 2, respectively. It can be seen that the energy of the IMFs of RPSEMD is equal to the simulation signal. The results demonstrate that it is feasible to use RPSEMD as the signal decomposition method in this paper.

### 4.2. Parameter Selection of DSE

Ship-radiated noise has obvious chaotic characteristics. In order to analyze the influence of DSE parameters on the entropy value, three typical chaotic signals are selected for simulation experiments. These chaotic signals can be expressed as follows [32]:(1)Henon mapping: $\left\{ \begin{array}{l} x(n+1)=1-1.4x{\left(n\right)}^{2}+y(n)\\ y(n+1)=0.3x(n) \end{array}\right.$, and the initial condition is $\left\{ \begin{array}{l} x_{0}=0\\ y_{0}=0 \end{array}\right.$. In this paper, we analyze the data points in the *y*-direction.(2)Rossler system: $\left\{ \begin{array}{l} \dot{x}=-(y+z)\\ \dot{y}=x+0.2y\\ \dot{z}=0.2+z(x-5) \end{array}\right.$, the initial condition is $\left\{ \begin{array}{l} x_{0}=-1\\ y_{0}=0\\ z_{0}=1 \end{array}\right.$, and the integral step size is 0.05. In this paper, we analyze the data points in the *x*-direction.(3)Mackey–Glass signal: $\frac{dx}{dt}=-bx(t)+\frac{ax(t-30)}{1+x^{10}(t-30)}$, where $\left\{ \begin{array}{l} a=0.2\\ b=0.1 \end{array}\right.$.

The original signals are taken from 5000 points of the above three types of simulation signals. The original signals are normalized to get the time-domain waveform shown in Figure 4a. We select different DSE parameters to calculate the entropy of simulated signals. In this section, we mainly study the influence of symbol length *m* and the adjustment factor *α* on the differential symbolic entropy. The symbol length is set from 1–5 with a step size of one, and the adjustment factor is set from 0.1–0.8 with a step size of 0.01. The DSE entropy under different parameters is shown in Figure 4b.

It can be seen from Figure 4b that when *m* is equal to one, the entropy of the three types of chaotic signals is closest. When *m* is greater than one, these signals can be better distinguished. When *m* is fixed, the value of *α* has no significant effect on the entropy of the Rossler signal and the Mackey–Glass signal. For the Henon mapping, as *α* increases, its entropy increases first and then decreases, and the maximum is obtained around 0.6. In conclusion, the value of *m* has little influence on DSE. When *α* is equal to 0.6, DSE can be used to classify the three types of chaotic signals better. Considering the calculation speed and the stability of DSE, we set *m* and *α* to three and 0.6, respectively.

It can also be seen from Figure 4b that the Henon mapping has the largest DSE entropy, which means it is more complex. The DSE of the Rossler signal are minimum, which means that the Rossler signal has regularity to some extent. The DSE of the Mackey–Glass signal are a bit lower than those of the Henon map, indicating that the Mackey–Glass signal also has higher complexity. The results demonstrate that DSE can discriminate the different simulation signals and can be used to quantify the information content of nonlinear time series.

To illustrate the influence of sampling points on entropy, we calculate DSE at different sampling points. Figure 5 shows the DSE under the different data points of these signals; when the number of sampling points is less than 500, the DSE fluctuates greatly; when the number of sampling points is greater than 500, the DSE is more stable. Therefore, when the number of sampling points of the signal is greater than 500, the result of DSE is reliable.

## 5. Analysis of Ship-Radiated Noise Based on RPSEMD, MI, and DSE

In this paper, all data of ship-radiated noise come from the official website of the National Park Service (available at http://www.nps.gov/glba/naturescience/soundclips.htm). Three different types of ship-radiated noise are selected as sample data, namely ferry, cruise ship, and freighter. For convenience, we named the three ship-radiated noise as Ship-I, Ship-II, and Ship-III, respectively. Each type of underwater acoustic signals has 30 sample data. Each sample length is 5000 points, and the sampling frequency is 44.1 kHz. The samples are normalized to get the time-domain waveform of three types of ship-radiated noise shown in Figure 6a, Figure 7a and Figure 8a, respectively. The abscissa represents the sampling point, and the ordinate represents the normalized amplitude. The RPSEMD decomposition results of the three types of ship-radiated noise are shown in Figure 6b, Figure 7b and Figure 8b, respectively.

Figure 9 shows the DSE of different sampling point numbers, whose entropy gradually stabilizes with the increase of the number of sampling points. When the number of sampling points reaches 500, the DSE entropy of the three types of ships-radiated noise are stable at around 0.9. It can be found that DSE is sensitive to noise, so these signals have a large entropy. Therefore, it is difficult to distinguish these three types of ships if the DSE of the original signal is directly used as the feature vector.

It can be seen from Figure 6b, Figure 7b and Figure 8b that the ship-radiated noise is decomposed into a series of IMFs by RPSEMD, and the frequency of the IMF decreases sequentially with the order of the modes. For different ship-radiated noise, the number of IMFs decomposed by RPSEMD is different. The norMI and DSE for each ship-radiated noise are listed in Figure 10.

In Figure 10, the abscissa represents the IMF ordinal number of ship-radiated noise, the red circle represents the norMI of IMF, and the blue triangle represents the DSE of IMF. It can be found that DSE decreases with the increase of IMF, indicating that the higher the order of IMF, the lower the complexity. Consequently, if only DSE is used as the feature vector, it is arduous to distinguish the three types of ship-radiated noise.

Li et al. [21] used the IMF with the highest energy as the principal IMF (PIMF) and used the permutation entropy of PIMF as the feature vector (namely EMD-PIMF-PE) to realize the classification of three types of ship-radiated noise. In Figure 10, the DSE of the IMF with the largest norMI is marked with a green dashed line, and the DSE of the IMF with the largest norMI is also listed in Table 3. It can be seen that for Ship-I and Ship-II, the DSE of the IMF with the largest norMI are around 0.27, and the values are quite close. Therefore, it may be arduous to distinguish between Ship-I and Ship-II using this method, which will be proven later.

Therefore, we consider using norMI as the weight, weighting the corresponding DSE and using the sum of the weighted DSE (WDSE) as the feature vector, giving the formula as follows:(19)$$WDSE=\sum_{i=1}^{K}norMI_{i}\cdot DSE_{i}$$
where $norMI_{i}$ represents the norMI of the *i*^th^ IMF, $DSE_{i}$ denotes the DSE of the *i*^th^ IMF, and *K* is the number of IMFs. The WDSE of the three ships’ radiated noise are listed in Table 4. It can be found that the difference between the feature vectors of Ship-I and Ship-II becomes larger.

## 6. Feature Extraction and Classification of Ship-Radiated Noise

### 6.1. Feature Extraction

To verify the effectiveness of the proposed feature extraction method, 30 samples of each type of ship-radiated noise are selected. The WDSE distribution of the three types of ship-radiated noise is shown in Figure 11a. The abscissa represents the number of samples, and the ordinate represents the feature vector WDSE. It can be seen that the WDSE of Ship-III is the largest, and the WDSE of Ship-II is the smallest. The results demonstrate that the WDSE value is at the same level for the same ships, but there is an obvious difference for different types of ships. The above results manifest that the proposed feature extraction method can distinguish three types of ship-radiated noise.

In order to prove the superiority of the proposed method, the DSE of original ship-radiated noise, the EMD-PIMF-PE method [21], and the DSE of the IMF with the largest norMI (IMF-norMI-DSE) are taken as the feature vector of ship-radiated noise, respectively. According to literature [21], the embedding dimension and time delay of permutation entropy are set as four and one, respectively. As shown in Figure 11b, due to the influence of the marine environment, the DSEs of the three types of ship-radiated noise are basically between 0.9 and 0.94. Therefore, it is not feasible to distinguish these ships directly by using DSE. It can be seen from Figure 11c that the EMD-PIMF-PE method can basically distinguish between Ship-II and Ship-III, but there is a large overlap between Ship-I and Ship-II. Figure 11c,d have similar results. It can be seen that the feature vector of Ship-III is the largest, and the feature vector of Ship-II is the smallest. This result is consistent with the proposed method in this paper. Compared with the proposed method, the IMF-norMI-DSE method is completely unable to identify Ship-I because their DSE values fluctuate greatly.

### 6.2. Classification

To realize the automatic identification of ship-radiated noise, the extracted features are input into the SVM for training and testing. For each type of ship-radiated noise, 10 samples are used as training samples, and the remaining 20 samples are used as test samples. To compare classification accuracy, the DSE of the original ship-radiated noise, the EMD-PIMF-PE method [21], and the IMF-norMI-DSE method are also used to classify ship-radiated noise. The SVM outputs of these four methods are shown in Figure 12, respectively, and the recognition rates are listed in Table 5. It can be found that the classification results of SVM are consistent with the feature extraction results in Section 6.1. For each type of ship-radiated noise, the DSE of the original signal is not completely classified correctly, and the classification accuracy is 48.3333%. The EMD-PIMF-PE method can classify Ship-II and Ship-III well, but cannot correctly identify Ship-I, and the classification accuracy is 70%. The IMF-norMI-DSE method is inferior to the EMD-PIMF-PE method, and the classification accuracy is 66.6667%. Compared with the other three methods, the classification accuracy of the proposed method reaches 98.3333%. The results indicate that the proposed method can better classify the three types of ship-radiated noise.

## 7. Conclusions

To improve the recognition accuracy of ship-radiated noise, a novel feature extraction method based on RPSEMD, MI, and DSE is proposed. The main findings in this paper are highlighted as follows:(1)A novel differential symbolic entropy for measuring the complexity of time series is introduced. DSE not only has the advantage of high computational efficiency, but also has a significant effect on shorter time series. It was first applied to underwater acoustic signal processing.(2)Simulation experiments demonstrate that RPSEMD can better alleviate the mode mixing problem compared with EMD and EEMD. Therefore, this paper uses RPSEMD as a signal decomposition tool.(3)Compared with [21], it is often the case that only one IMF with the principal features is selected for feature extraction. In this paper, the entropy is weighted by norMI, so the importance of each IMF is considered.(4)The method proposed in this paper can extract the characteristics of ship-radiated noise more precisely and comprehensively, and the classification accuracy reaches 98.3333%.

Last but not least, this paper provides a new idea for feature extraction of nonlinear and non-stationary signals. For example, the weight coefficient can be replaced by the IMF’s energy and other parameters. It is worth noting that although both EEMD and RPSEMD improve the mode mixing problem of EMD, RPSEMD is more efficient than EEMD and therefore has better utility. In the following work, we will use RPSEMD to reduce noise of underwater acoustic signal and compare it with EEMD.

## Figures and Tables

**Figure 1 entropy-21-00176-f001:**
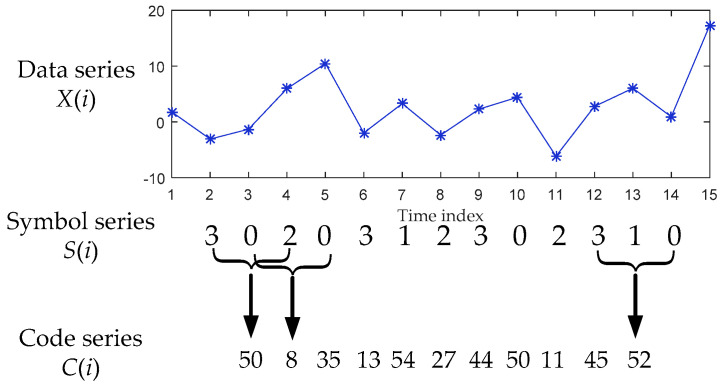
Process of symbolization and coding (data and symbols in virtual frame will not be symbolized or encoded). In the creation of code series, symbol length *m* is three. The first and last elements will not be transformed according to the determination of symbolization, and the last *n* − 1 bit symbols are not encoded for this encoding process.

**Figure 2 entropy-21-00176-f002:**
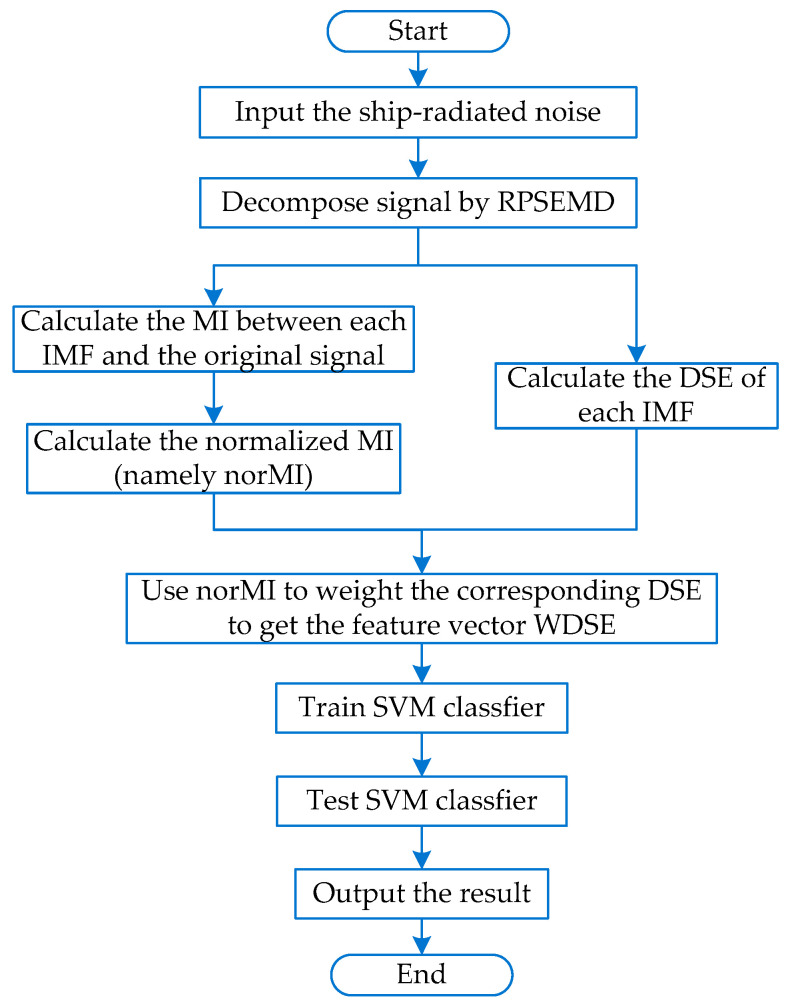
The flowchart of the proposed method. RPSEMD, regenerated phase-shifted sinusoid-assisted empirical mode decomposition; IMF, intrinsic mode function; WDSE, weighted differential symbolic entropy.

**Figure 3 entropy-21-00176-f003:**
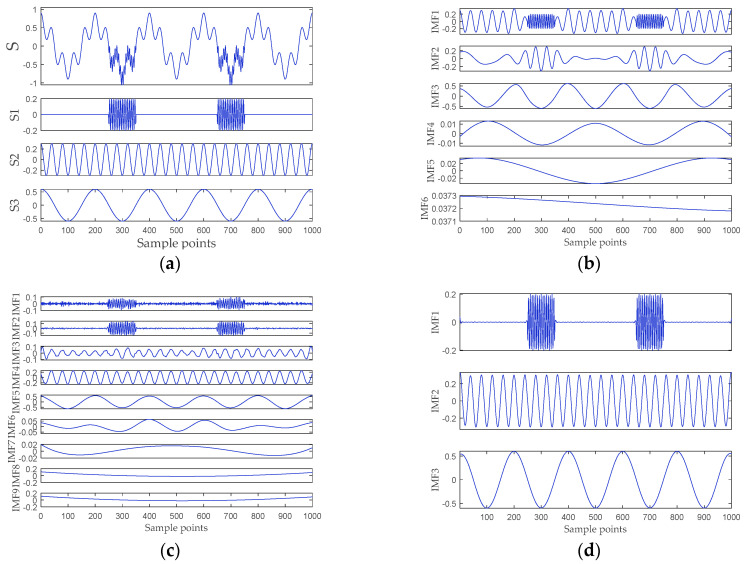
The simulation signals and the decomposition result of EMD, ensemble EMD (EEMD), and RPSEMD. (**a**) The simulation signals; (**b**) the decomposition result of EMD; (**c**) the decomposition result of EEMD; and (**d**) the decomposition result of RPSEMD.

**Figure 4 entropy-21-00176-f004:**
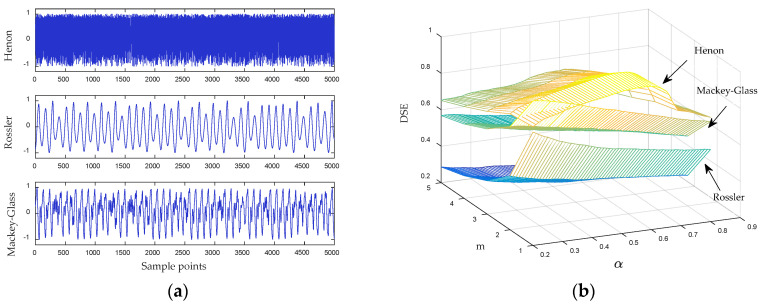
(**a**) The normalized simulation signals and (**b**) DSE values of three types of simulated signals under different parameters.

**Figure 5 entropy-21-00176-f005:**
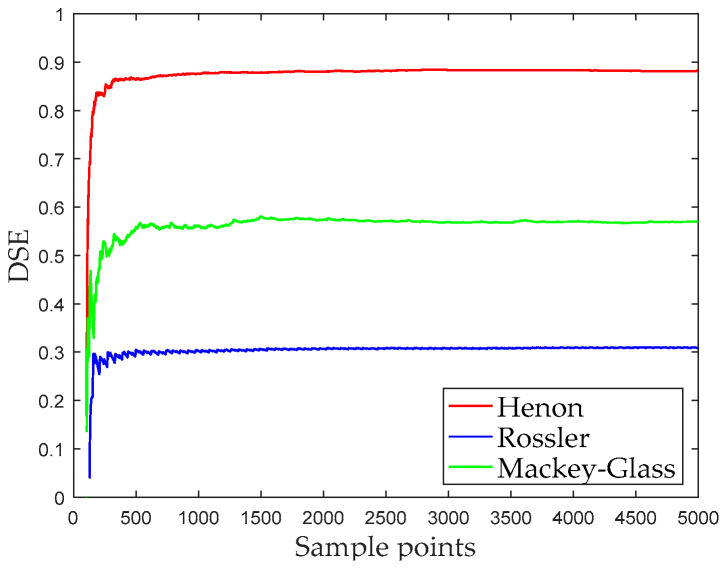
The DSE under the different data points of four simulation signals.

**Figure 6 entropy-21-00176-f006:**
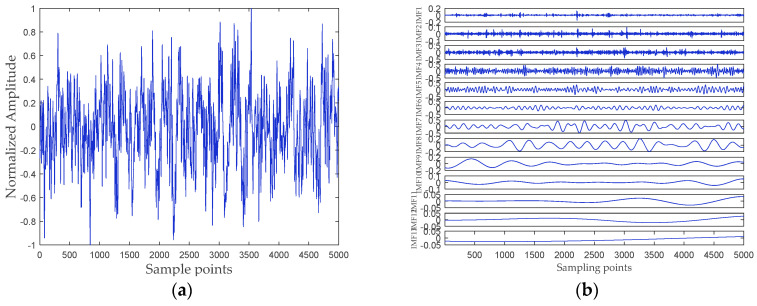
(**a**) The time-domain waveform of Ship-I; and (**b**) the decomposition result of RPSEMD.

**Figure 7 entropy-21-00176-f007:**
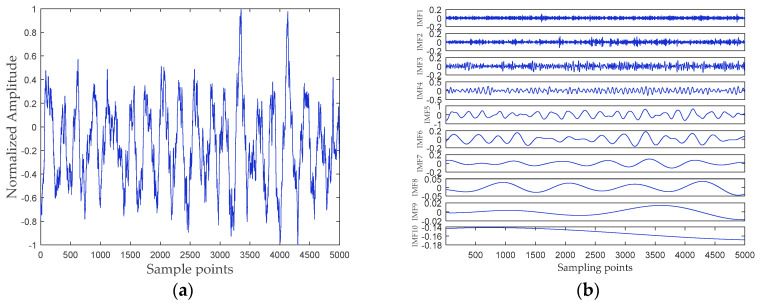
(**a**) The time-domain waveform of Ship-II; and (**b**) the decomposition result of RPSEMD.

**Figure 8 entropy-21-00176-f008:**
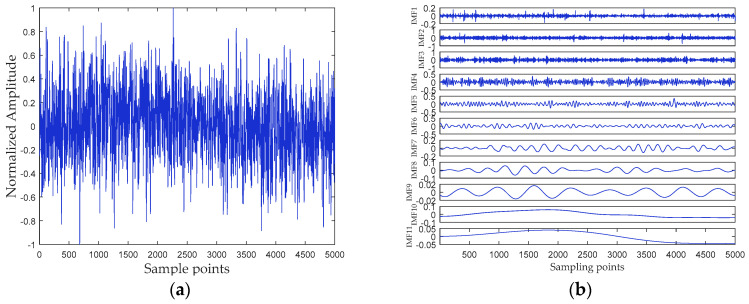
(**a**) The time-domain waveform of Ship-III; and (**b**) the decomposition result of RPSEMD.

**Figure 9 entropy-21-00176-f009:**
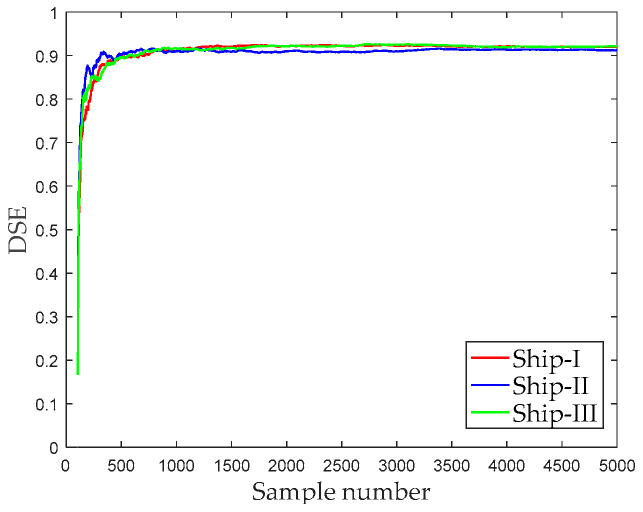
The influence of sampling point number on DSE.

**Figure 10 entropy-21-00176-f010:**
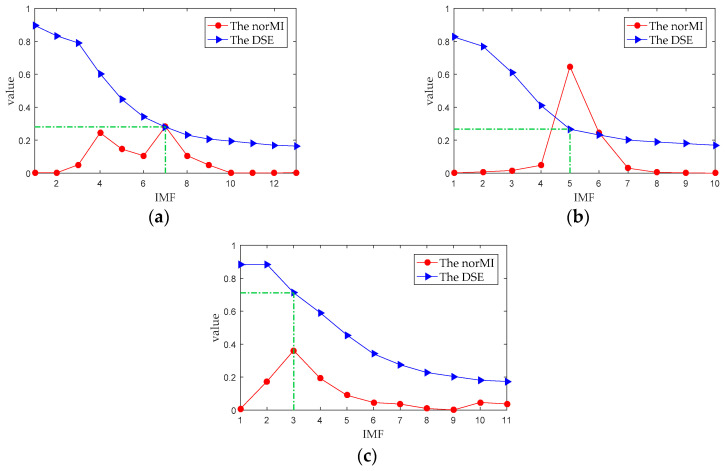
The norMI and DSE of the IMFs. (**a**) Ship-I; (**b**) Ship-II; and (**c**) Ship-III.

**Figure 11 entropy-21-00176-f011:**
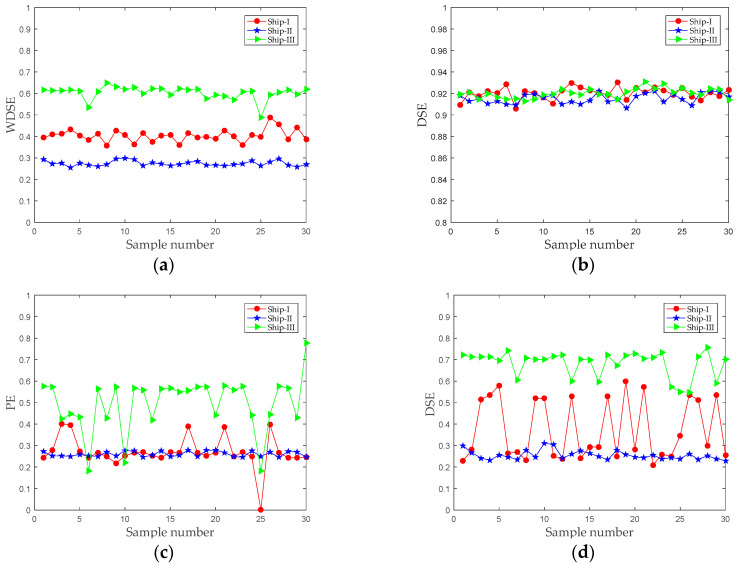
Feature extraction results of different methods. (**a**) The proposed method; (**b**) The DSE of original ship-radiated noise; (**c**) The EMD-PIMF-PE method; and (**d**) The IMF-norMI-DSE method.

**Figure 12 entropy-21-00176-f012:**
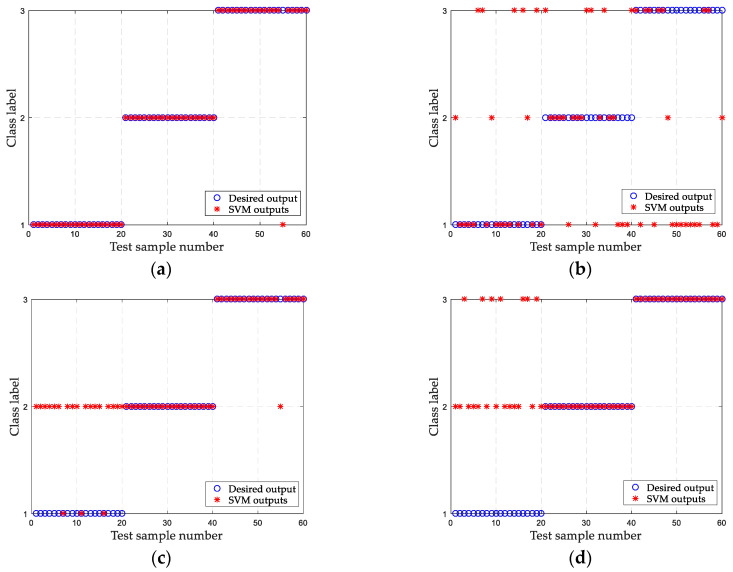
SVM classification results of different methods. (**a**) The proposed method; (**b**) the DSE of original ship-radiated noise; (**c**) the EMD-PIMF-PE method; and (**d**) the IMF-norMI-DSE method.

**Table 1 entropy-21-00176-t001:** The energy of simulation signals.

s1	s2	s3
3.9600	45.0000	180.0000

**Table 2 entropy-21-00176-t002:** The energy of IMFs.

Method	IMF1	IMF2	IMF3	IMF4	IMF5	IMF6	IMF7	IMF8	IMF9
EMD	38.0397	15.9000	158.5438	0.0703	0.6678	1.3866	/	/	/
EEMD	0.5846	1.8154	1.8405	28.9382	163.4087	0.7825	0.1020	0.0258	1.7101
RPSEMD	3.9116	45.0832	177.7195	/	/	/	/	/	/

**Table 3 entropy-21-00176-t003:** The DSE of PIMF for three ships’ radiated noise.

Parameter	Ship-I	Ship-II	Ship-III
The index of the IMF with the largest norMI	7	5	3
The DSE of the IMF with the largest norMI	0.2802	0.2664	0.7116

**Table 4 entropy-21-00176-t004:** The WDSE of three ships’ radiated noise.

Parameter	Ship-I	Ship-II	Ship-III
WDSE	0.3939	0.2931	0.6154

**Table 5 entropy-21-00176-t005:** SVM classification results of different methods.

Methods	Accuracy Rate
The proposed method	98.3333%
The DSE of original ship-radiated noise	48.3333%
The EMD-PIMF-PE method [6]	70%
The IMF-norMI-DSE method	66.6667%

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
