# Peer review of "Feature Extraction of Ship-Radiated Noise Based on Regenerated Phase-Shifted Sinusoid-Assisted EMD, Mutual Information, and Differential Symbolic Entropy"

_entropy, 2019, doi:10.3390/e21020176_

Round 1

Reviewer 1 Report

The paper authored by G. Li et al. is focused on the recognition of ship-radiated noise. The authors propose a novel method for performing this task. It turns out that the method is, in fact, a well-engineered scheme that has three important ingredients. The work is interesting, especially because of the impressive accuracy rate reported in Table 5 (for SVM classification). However, the paper is hard to read and it is difficult for the readers to implement themselves the newly proposed method. For each step of the algorithm, the user should select some parameters. It might be very useful to provide a table in which to list all these parameters and their values. I am not able to understand from where can be downloaded the code used in the experiments. In my opinion, the authors should make the code publicly available.

Not all the equations are properly written. For example, I assume that the last row in Eq. (4) should be $3:diff \leq -\alpha var$. From the definition of the time series X (page 4, row 138) is not obvious if the time series is multivariate or univariate. The definition of n on page 4 (row 147) is not very clear. In equations (10)-(13), s2 and s3 are defined for all n in the interval (0,1). However, s1 is defined only when n belongs to two subintervals of (0,1). Should we assume that s1=0 outside these two subintervals? The authors should provide references for the identities in the equations (7) and (8). The structure of Section 5 is weird.

Author Response

Attached is the response of review.

Reviewer 2 Report

The authors presents a methodology using empirical mode decomposition (RPSEMD), mutual information (MI) and differential symbol entropy (DSE) to improve the recognition accuracy of ship-radiated noise. This reviewer has the following comments:

You mention in Page 2 that few studies have applied RPSEMD and DSE to underwater acoustic signal processing. Please cite these studies. Then in section of result I expect a comparison with these studies.

Although is mention the advantage of differential symbol entropy (DSE), there is not a feature selection step to probe if is the best feature.

I think the paper would improve if there is a results comparison using different features and EEMD using the same methodology proposed. Add discussion section of this comparison.

Author Response

Attached is the response of review.

Reviewer 3 Report

January 15, 2019

Review of the paper entitled
Feature Extraction of Ship-Radiated Noise Based on Regenerated Phase-Shifted Sinusoid-Assisted EMD, Mutual Information, and Differential Symbolic Entropy
by Guohui Li, Zhichao Yang and Hong Yang

Manuscript ID: entropy-429217
In this paper, the Authors proposed a feature extraction method to improve the recognition accuracy of ship-radiated noise. This approach is based on regenerated phase-shifted sinusoid-assisted empirical mode decomposition (RPSEMD), mutual information (MI) and differential symbol entropy (DSE). First, using RPSEMD the ship-radiated noise is decomposed into a series of intrinsic mode functions (IMFs). Next, the MI between each IMF and the original signal is calculated. Then, each normalized MI (norMI) is used as the weight coefficient to weight the corresponding DSE, and the weighted DSE (WDSE) is obtained. Finally, the WDSEs are sent into the support vector machine (SVM) classifier to classify and recognize three types of ship-radiated noise. Recognition rate of the proposed method reaches 98.3333%.

Unfortunately, in my opinion the study proposed suffers essential imperfections:

1.The Authors present in a very compact form a number of concepts like: phase-shifted sinusoid-assisted empirical mode decomposition, mutual information and differential symbol entropy. Most descriptions are presented in not precise form. Terms in the formulas should be defined more precisely from mathematical point of view.  
2. The Conclusion Section is to short and should stresses the added value of the paper. What follows from result obtained.
3.There are at least a couple of references form prestigious journals as well as last papers that are important and are missing.
For example:
Wales, SC; Heitmeyer, RM, An ensemble source spectra model for merchant ship-radiated noise,
Journal of the Acoustical Society of America111(3), 1211-1231, 2002
Bao, Fei; Li, Chen; Wang, Xinlong; et al., Ship classification using nonlinear features of radiated sound: An approach based on empirical mode decomposition, Journal of the Acoustical Society of America 128(1), 206-214, 2010
Bao, Fei; Wang, Xinlong; Tao, Zhiyong; et al., EMD-based extraction of modulated cavitation noise
Mechanical Systems and Signal Processing 24(7), 2124-2136, OCT 2010
Knobles, D. P., Maximum entropy inference of seabed attenuation parameters using ship radiated broadband noise, Journal of the Acoustical Society of America 138(6), 3563-3575,  2015
Firat, Umut; Akgul, Tayfun, Compressive Sensing for Detecting Ships With Second-Order Cyclostationary Signatures, IEEE Journal of Oceanic Engineering 43(4), 1086-1098, 2018
4. There are some repetitions. Please remove them.
5.There are some typos and grammatical writing issues, please have a very careful check of the writing.

The Authors have experience in recognition accuracy of ship-radiated noise. The idea presented in the paper may be interesting for broad audience and it is promising, but at this moment the manuscript, due to the weaknesses listed above, requires at least major revision.

Author Response

Attached is the response of review.

Round 2

Reviewer 2 Report

No comments. The authors made the requested improvements.

Reviewer 3 Report

I would consider this paper for publication.